# Integrative In Silico and In Vitro Transcriptomics Analysis Revealed Gene Expression Changes and Oncogenic Features of Normal Cholangiocytes after Chronic Alcohol Exposure

**DOI:** 10.3390/ijms20235987

**Published:** 2019-11-28

**Authors:** Suthipong Chujan, Tawit Suriyo, Jutamaad Satayavivad

**Affiliations:** 1Applied Biological Sciences Program, Chulabhorn Graduate Institute, Chulabhorn Royal Academy, Bangkok 10210, Thailand; 15020106@cgi.ac.th; 2Laboratory of Pharmacology, Chulabhorn Research Institute, Bangkok 10210, Thailand; tawit@cri.or.th; 3Center of Excellence on Environmental Health and Toxicology, Office of Higher Education Commission, Ministry of Education, Bangkok 10400, Thailand

**Keywords:** cholangiocarcinoma, chronic alcohol exposure, normal cholangiocyte, oncogenic feature, transcriptomic analysis

## Abstract

Cholangiocarcinoma (CCA) is a malignant tumor originating from cholangiocyte. Prolonged alcohol consumption has been suggested as a possible risk factor for CCA, but there is no information about alcohol’s mechanisms in cholangiocyte. This study was designed to investigate global transcriptional alterations through RNA-sequencing by using chronic alcohol exposure (20 mM for 2 months) in normal human cholangiocyte MMNK-1 cells. To observe the association of alcohol induced CCA pathogenesis, we combined differentially expressed genes (DEGs) with computational bioinformatics of CCA by using publicly gene expression omnibus (GEO) datasets. For biological function analysis, Gene ontology (GO) analysis showed biological process and molecular function related to regulation of transcription from RNA polymerase II promoter, while cellular component linked to the nucleoplasm. KEGG pathway presented pathways in cancer that were significantly enriched. From KEGG result, we further examined the oncogenic features resulting in chronic alcohol exposure, enhanced proliferation, and migration through CCND-1 and MMP-2 up-regulation, respectively. Finally, combined DEGs were validated in clinical data including TCGA and immunohistochemistry from HPA database, demonstrating that *FOS* up-regulation was related to CCA pathogenesis. This study is the first providing more information and molecular mechanisms about global transcriptome alterations and oncogenic enhancement of chronic alcohol exposure in normal cholangiocytes.

## 1. Introduction

Biliary tract cancer (BTC) or cholangiocarcinomas (CCA) are malignancy tumors arising from biliary epithelia. They are categorized into intrahepatic (iCCA), perihilar (pCCA), and distal (dCCA) cholangiocarcinoma. The epithelial malignancies originate from transformed cholangiocytes [1]. These cancers present in some specific areas with specific possible risk factors such as chronic cholangitis, liver fluke infection, and alcohol consumption [2].

Alcohol has been classified as a class I carcinogen for several types of cancers [3]. Excessive chronic alcohol consumption is a widely acknowledged possible risk factor for CCA. The carcinogenic effect of alcohol exposure on the cholangiocyte is sophisticated due to its effects on cholesterol metabolism, leading to a reduction of gallstone formation [4,5]. It has been shown that alcohol may inhibit DNA methylation, interaction with retinoid metabolism, and generation of reactive oxygen species (ROS), of which reactive free radicals from alcohol metabolism have the ability to form adducts with DNA, causing oxidative DNA damage [6]. The one proposed mechanism of the putative oncogenic features of alcohol involves stimulation of cell proliferation and migration, which mediates increasing cyclin D 1 (CCND1) and matrix metalloproteinase 2 (MMP2) expression, respectively [7]. Although several studies have reported positive associations between alcohol consumption and various cancers, the carcinogenicity of alcohol in epithelia of the biliary tract is still unclear [8].

Next generation sequencing (NGS) techniques will certainly become a gold standard for understanding the molecular mechanisms of toxicity. RNA sequencing generates deep sequencing data for the direct quantification of transcripts. It also provides large expression data for comprehensive bioinformatics identification [9]. Bioinformatics and computational analysis have been usefully applied in studies of various cancers and have been confirmed to provide efficient and reliable biomarkers for cancer diagnosis and therapeutic targets [10]. In this study, integrative transcriptome analysis was designed to characterize all transcriptional activity in immortalized human cholangiocyte MMNK-1 cell line with chronic alcohol exposure. To study the association between chronic alcohol exposure and cholangiocarcinogenesis, the Differentially Expressed Genes (DEGs) from in vitro RNA-Seq were, through the use of bioinformatics analysis, merged with the DEGs of in silico CCA transcriptomics Gene Expression Omnibus (GEO) datasets.

The aims of the present study were to investigate (a) the transcriptomics alteration profile triggered by chronic alcohol exposure associated with CCA, (b) the early molecular biomarker associated with pathogenesis in CCA patients by using expression profile database, and (c) the possible oncogenic features of chronic alcohol exposure in immortalized normal human cholangiocytes.

## 2. Results

### 2.1. Cell Viability

To examine the toxic effects of chronic alcohol in MMNK-1 cells, cells were treated with several doses of alcohol at 10, 20, 50, 80, 100, 500, and 1000 mM for 7 days. Cytotoxicity was observed by MTT assay. Cell viability examination suggested that alcohol at 500 and 1000 mM can inhibit cell proliferation ~50%. No cell toxicity was observed at 20 mM, as shown in Figure 1. In addition, there was no change in cell morphology after 7 days of 20 mM alcohol treatment. Therefore, the 20 mM was used for chronic alcohol exposure in MMNK-1.

### 2.2. RNA Extraction, Sequencing and Quantification

RNA was isolated from un-treated and chronic 20 mM alcohol-treated cells for RNA sequencing analysis. Data acquisition that composed of obtaining raw read, read alignment, and quantification, was quality checked at each step. FastQC version 0.3 was used to calculate for quality checking and showed the low error rate of 0.1%. The percentage of mapped reads indicated high overall sequence accuracy and low DNA contamination. The RNA integrity number (RIN) score was above 9.0, and rRNA ratio (28S/18S) was above 1.9, indicating that the obtained RNA was high quality nucleic acid.

### 2.3. Gene Expression Profile and Differentially Expressed Genes (DEGs) Identification of In Vitro and In Silico

For in silico meta-analysis, we integrated three GEO datasets (GSE31370, GSE32879 and 32225) including 18 normal and 171 CCA patients by using Limma R package. Quality control, based on the percentage of missing value, was performed for each dataset. The boxplot showed the centrality measure of each dataset. These plots showed homogeneity in the expression values. Under the threshold FDR < 0.05 and log2 fold change ≥ 2, a total of 4381 genes were identified, including 1821 down- and 2560 up-regulated genes which were normal, compared with CCA. The DEGs expression hierarchical clustering heat maps (overall and top 100 up- and down-regulated genes) are presented in Figure 2 and Appendix A.

Based on RNA-sequencing, the results from DESeq2 analysis was further analyzed to calculate genes with significant differential expression according to the criteria of log2 fold change greater than 0.4 and *p*-value less than 0.05. The number of up- and down-regulated genes were 57 and 62, respectively. Clustering analysis was calculated and data classified by similarity. Samples or genes with similar expression patterns were grouped. The FPKM value of different genes under different experimental conditions was taken as the expression level and was used for hierarchical clustering heat map, as shown in Figure 3 and Appendix A.

The DEGs from in silico and RNA-seq were combined with a Venn diagram using Venny 2.1.0 online tool (https://bioinfogp.cnb.csic.es/tools/venny/). As shown in Figure 4, we found 19 overlapping DEGs existing in both the CCA and the alcohol-treated groups. The gene list and description are shown in Table 1.

### 2.4. Functional and Pathway Enriched Analysis.

To obtain insight into the biological function of the combined DEGs, they were applied to the DAVID database. The results of the top 10 GO analyses demonstrated that the positive regulation of transcription from RNA polymerase II promoter, as well as transcription from RNA polymerase promoter and nucleoplasm were significantly enriched in Biological process (BP), Molecular function (MF) and Cellular component (CC), respectively. On the other hand, we further studied on the signaling pathway enrichment by using KEGG database. The results showed that pathways in cancer were significantly enriched pathways. The GO and KEGG enrichment are listed in Figure 5.

### 2.5. Protein-Protein Network and Hub-Gene Identification

Hub nodes potentially have a key role in signaling. Strong interactions with dysregulated genes usually mean integrated multiple downstream signaling. Hub-genes identification made protein-protein network interaction analysis of 19 overlapping DEGs possible. NetworkAnalyst database provided a protein network with 1058 nodes, 1226 edges, and 17 seeds. Twelve nodes with degrees above 10 were identified as hub genes, including Early Growth Response 1 (*EGR1*), Fos Proto-Oncogene (*FOS*), Tubulin Beta Class I (*TUBB*), CCAAT Enhancer Binding Protein Beta (*CEBPB*), Four and A Half LIM Domains 1 (*FHL1*), Colony Stimulating Factor 2 (*CSF2*), Tripartite Motif Containing 38 (*TRIM38*), Iodothyronine Deiodinase 2 (*DIO2*), Follistatin (*FST*), Transient Receptor Potential Cation Channel Subfamily C Member 4 (*TRPC4*), Melanocortin 1 receptor (*MC1R*) and Ubiquitin C (*UBC*) as shown in Figure 6 and Table 2.

### 2.6. TCGA Verification

In order to explore the combined 19 DEGs, we assayed the TCGA database for CCA patient (TCGA provisional) with the cBioportal. Analyzing the mRNA data for 19 DEGs, *DBH,* and *FOS* expression were found to be significantly overexpressed (*p* < 0.05) with the altered group. The volcano plot and box plot are presented in Figure 7.

### 2.7. Immunohistochemistry Verification in Human Protein Atlas

The immunostaining level of 19 combined DEGs were validated in human protein atlas database as shown in Figure 8. According to the results, *FOS*, *IFNGR2*, *CEBPB*, and *ZNF615* expressions were found to be significantly different among normal cholangiocyte and CCA tissues.

### 2.8. Chronic Alcohol Exposure Increased Proliferation in MMNK-1 Cell

To understand the proliferation effects of chronic alcohol exposure on MMNK-1, the doubling time of MMNK-1 after long-term chronic treatment with and without alcohol were determined. Notably, there was no change in cell morphology after 60 days of 20 mM alcohol treatment. The cell numbers at many time points (0, 1, 2, 3, 4, 5 and 6 days) were assessed by using cell counter and trypan blue dye-exclusion, as shown in Figure 9. Chronic alcohol treatment showed a significant reduction in the doubling time of MMNK-1 from 1.855 day to 1.269 day. CCND1 plays an important role in cell proliferation and tumorigenesis. We next observed CCND1 expression levels in alcohol-treated MMNK-1, using western blot analysis (Appendix A). The results demonstrated that the CCND1 expression levels of the alcohol treated group significantly increased, compared to the untreated group. In terms of cell doubling time and CCND1 expression, our study provides some support for the possibility that chronic alcohol exposure induces cell proliferation and up-regulated CCND1 expression.

### 2.9. Chronic Alcohol Exposure Enhanced the Migration Activity of MMNK-1 Cells

To examine the effects of chronic alcohol exposure on MMNK-1 migration, the migration activity was observed at 0, 24 and 48 h. The results demonstrated that alcohol treated group significantly accelerated the migration activity of MMNK-1 cells. The quantification of wound area showed that at 24 h. the wound area ~20% compared to the control group ~59% and after 48 h. the wound area ~4% compared to the control ~31% as shown in Figure 10. The expression of matrix metalloproteinase-2 (MMP-2) has an important role for extracellular matrix degradation that involved in the cells motility process. We further assessed the alcohol stimulated MMNK-1 in expression of migration-linked MMP-2. As presented in Figure 10, the results showed increased MMP-2 expression, compared to the untreated group (Appendix A). Our studies indicated that chronic alcohol exposure could enhance MMP-2 expression and cell migration of MMNK-1.

## 3. Discussion

The negative lifestyle for CCA could be consumption of raw freshwater fish infected with liver fluke *Opisthorchis viverrini*, beef sausage, alcohol, and tobacco use. These established possible risk factors for the CCA are very heterogeneous. The mutational landscape, the cellular origins and the carcinogenesis of the different CCA subtypes are still unclear [11]. The current knowledge of alcohol toxicity causing several cancers is well known. Here, as shown in Figure 11, we examined the effects of prolonged exposure to alcohol on human normal cholangiocytes and extracted out the total RNA for RNA sequencing as in vitro transcriptomics model. On the other hand, we performed an in silico computational bioinformatics analysis between healthy subjects and CCA patients by using GEO datasets from our selection criteria. To observe low alcohol induces the development of CCA, the DEGs from in vitro and in silico were merged and presented in the combined DEGs. We further analyzed the GO and KEGG pathways for identification of biological function by using combined DEGs. The results demonstrated that pathways involving in cancer were the most significant pathways. From KEGG results, oncogenic features including proliferation and migration were proved by our in vitro experiments. In addition, the transcriptomics alterations presented some genes that were related to clinical data with CCA. The candidate genes were validated by using TCGA and HPA and could be used as potential biomarkers for early CCA. This study is the first report of a transcriptomics analysis on normal human cholangiocytes involving continuous exposure for 2 months with low concentration of alcohol (20 mM). We found that even relatively low doses of alcohol contribute (a) alterations in global transcription activity, (b) the oncogenic features, including proliferation and migration mediated through CCND-1 and MMP-2 expression and (c) a potential early biomarker for CCA patients.

In silico and in vitro transcriptome data could offer accurate genome and transcription information especially in revealing affected biological pathways and processes that have previously not been considered. We identified 119 genes from in vitro (RNA-Seq) and 4381 genes from in silico experiments. There was a combined showing total of 19 genes. The enormous changes reflect not only cancer induction, but also many genes that are involved in the infection, transcription factor induction, and inflammatory responses. This is similar to previous reports showing that alcohol triggers oncogenicity, inflammation induction, and susceptibility to infection.

The biological process of GO analysis showed that the regulation of transcription from RNA polymerase II promoter was most significant. The RNA polymerase II core promoter is defined as a sequence that controls transcription initiation and epigenetics regulation. Previous reports have pointed out that chronic alcohol exposure in neurons resulted in the decrease CpG island methylation, resulting in reduced DNA methylation and relaxation of nucleosome packing [12]. KEGG analysis has shown that pathways in cancer were the most enriched, and that they were followed by infection and inflammatory regulation. To our best knowledge, chronic alcohol consumption is an important risk factor in the development of different types of cancers. We also extended the oncogenic features investigation to prove the hypothesis from our KEGG analysis that even low concentrations of alcohol in repeated exposure stimulates oncogenesis. Similar to previous in vitro studies, 25 mM ethanol in MCF-12A cells induced growth and oncogenic effects and also caused transcriptional signature alterations [13]. Moreover, multiple mechanisms have been identified as underlying the immunosuppressive effects of alcohol. These include a host of defensive systems in the respiratory and gastrointestinal tract, as well as all of the essential components of the immune system which is compromised by both direct effects of alcohol and alcohol-related dysregulation [14]. Previous reports have demonstrated that, chronic alcohol exposure in utero interferes with normal T-cell and B-cell development which may increase the risk of multiple infections during childhood and adulthood. Alcohol’s impact on T cells and B cells also increases the risk of infections such as pneumonia, HIV infection, hepatitis C viral infections and tuberculosis. It also impairs responses to vaccinations against infections, exacerbates cancer risk, and interferes with delayed-type hypersensitivity [15]. However, the molecular mechanisms underlying ethanol negative impact on the immune system remain poorly understood and further in vivo study is needed to prove this concept.

In a previous study, long-term alcohol exposure in low concentrations showed oncogenic features and global transcriptional changes in normal breast epithelial cells and normal human pancreatic ductal epithelial cells [13,16]. According to our study, results showed that the MMNK-1 cells with chronic alcohol exposure reduced their doubling time. They exhibited a strong proliferative effect and mediated through CCND-1 up-regulation. Long-term alcohol exposure also showed an induction of cell migration and cell motility enhancement effects via MMP-2 expression. CCND-1 interacts with cyclin-dependent kinase 4 (CDK4) or CDK6 to induce cell cycle progression from the G1 to S phase, promoting cell division or transformation. Aberrant expression of CCND-1 may cause an imbalance of the cell cycle, resulting in tumorigenesis [17]. High expression of matrix metalloproteinase-2 (MMP-2) was found to be correlated with tumor progression and poor prognosis in a variety of carcinomas. In a previous study, up-regulations of MMP-2 were found to be a marker for increased tumorigenesis of human intrahepatic CCA [18]. Invasion assay should be performed to support the effect of alcohol on MMP-2-associated invasiveness induction. However, our data suggests that low dose, chronic alcohol exposure contributes to oncogenic features that tend toward the development of cholangiocarcinoma. In addition, the expression of CCND1 and MMP-2 could be used as possible candidate molecular markers for early alcohol enhanced cell transformation. However, further studies should be done to confirm their contributions.

With protein-protein network interaction, the most 10 hub-genes were identified. They showed that Early growth factor-1 (*EGR*-*1*) was the highest centrality. *EGR*-*1* (also known as nerve growth factor induced-A) is a zinc-finger transcription factor responsible for regulation of cell growth and proliferation. *EGR*-*1*, an immediate early gene, is rapidly and transiently induced. It responds to various stimuli, including cytokines and growth factors, as well as to environmental stress, chemical exposure and tissue damage [19]. Our results demonstrate that *EGR*-*1* mRNA was up-regulated after prolonged alcohol exposure to human normal cholangiocytes (Appendix A). Similar previous reports indicate that early-life exposure to arsenic causes hypomethylation of *EGR*-*1* promoter and also increases mRNA expression [20]. *EGR*-*1* overexpression links to alcohol-induced liver disease and related to CCA may be used as a potential target for alcoholic liver disease diagnosis [21]. Interestingly, *EGR*-*1* alterations may have a key role in responsive alcohol exposure mechanisms, which provides an important strategy for the development of a new molecular therapy for the treatment of CCA.

To observe the association of alcohol-induced CCA, we validated the 19 combined genes in The Cancer Genome Atlas (TCGA) database. These efforts made it clear that mRNA expression of *DBH* and *FOS* are significantly associated and may play a crucial role in the aggressiveness of CCA. Additionally, we also verified combined genes in the Human Protein Atlas (HPA) database, which identifies of candidates for relevant biomarkers. Our immunohistochemistry results showed that *FOS*, *INFGR2*, *CEBPB*, and *ANF615* expression are associated with the pathogenesis of CCA. *FOS* is surprisingly associated with CCA pathogenesis in both databases. *FOS* (*c*-*fos*) is an important proto-oncogene that has been shown encoding for growth factors, cell surface receptors, membrane proteins, phosphokinases, and nuclear proteins. The *c*-*Fos* protein forms a heterodimer complex. With the transcription factor *c*-*Jun*/*AP*-*1* and *Fos*/*Jun* complex, *c*-*Fos* has the ability to bind to the regulatory elements of other genes. Chronic alcohol consumption leads to decrease retinoic acid by acceleration of retinoic acid metabolism, resulting in overexpression of the Activator protein 1 (*AP1*) gene associated with an increase in their proteins *c*-*jun* and *c*-*fos* [22]. Eventually, alcohol leads to increased CCND-1 expression, which is associated with hyperproliferation in the liver [23]. Thus, retinoic acid deficiency due to chronic alcohol exposure is associated with acceleration of carcinogenesis [24]. In a previous in vivo study, *c*-*Fos* overexpression appeared to be associated with high cellular proliferating activity and cell transformation in hamster cholangiocarcinogenesis [25]. In addition, a previous report showed that decreased expression levels of *IFNGR2* are a risk factor for tumorigenesis in humans and may involve IFN-γ dependent cancer immunosurveillance [26]. Meanwhile, there is no current publication which shows the important role of *DBH*, *CEBPB* and *ZNF615* in CCA. However, the *FOS* expression may be used as a potential early biomarker in alcohol induced CCA. However, further experimental in vivo studies are needed to verify our in vitro and in silico results.

## 4. Materials and Methods

### 4.1. Selection and Identification of Gene Expression Dataset for Meta-Analysis

Three microarray datasets available on Pubmed database were selected according to our selection criteria. The key word used was “cholangiocarcinoma”. Gene Expression Omnibus (GEO): GSE31370, GSE32879, and GSE32225 were analyzed in this study. Information was extracted from each study, including accession number, disease, microarray platform, number of cases, and controls and references. Inclusion criteria were set and strictly identified for selection of dataset, i.e., human case and control study, gene expression profile, comparative condition, and complete raw data as shown in Figure 12. Quality controls from three studies were performed with R package, using ImaGEO online tool [27]. The results showed the distribution of expression values in boxplots and the missing values for each dataset.

### 4.2. DEGs Identification from In Silico Bioinformatics Meta-Analysis

Three microarray data sets of the diseases were selected for gene expression analysis using ImaGEO, a web interface, to integrate and perform meta-analysis. Data was retrieved and loaded by GEO IDs and processed by using GEO query package. As shown in Table 3, meta-analysis containing three studies were performed in Meta DE R package; they were effect size (EF), Fisher’s test, and adjusted *p*-value. The results are displayed in an interactive table with the significant genes, gene symbol, gene name, *p*-value, adjusted *p*-value, and fold change value.

### 4.3. Cell Culture and Treatment

Immortalized normal human cholangiocyte MMNK-1 cells were purchased from the Japan Collection of Research Bioscience (JCRB) cell bank, Osaka, Japan. MMNK-1 cells were cultured in Dulbecco’s Modified Eagle’s Medium (DMEM) containing 10% fetal bovine serum (JR scientific, Inc, Woodland, CA, USA), 2 mM l-Glutamine, 100 U/mL penicillin and 100 µg/mL streptomycin (Gibco, Carlsbad, CA, USA), and maintained at 37 °C in 5% CO_2_ humidified atmosphere. For alcohol exposure, MMNK-1 cells were treated with alcohol in fresh medium 24 h after initial seeding. We chose alcohol treatment concentration at 20 mM to represent chronic alcohol exposure, as a previous study described where the chronic alcohol treatment at 20 mM for nine days could alter cellular functionality of rat astrocyte cell culture [28]. For information, the 20 mM in vitro alcohol concentration approximates a 100 mg/dL blood alcohol level, which is achieved in vivo after a dose of moderate drink and is the upper range of blood alcohol concentration in most U.S. jurisdictions [29]. The media with ethanol were replaced every 24 h with fresh media containing the 20 mM ethanol concentration. The cells were treated for 60 days (2 months) and trypsinized 80% twice weekly when archived. The duration of long-term alcohol exposure for 60 days was referred from previous established long-term alcohol exposure model [30].

### 4.4. Cell Viability Assay

Cell viability assay was measured by a quantitative colorimetric assay (MTT) (1-(4,5-Dimethylthiazol-2-yl)-3,5 diphenyl (formazan) (Sigma-Aldrich, St. Louis, MO, USA) showing the mitochondrial activity of living cells. MMNK-1 cells were seeded in 96-well plates (1 × 10^3^ cells/well) and cultured overnight for cell attachment. The cells were exposed to 0, 10, 20, 50, 80, 100, 500, and 1000 mM ethanol for seven days. The medium, with and without ethanol, was changed every day. At the end of incubation time, the medium was removed. MTT stock solution was prepared as 5 mg/mL in phosphate buffer saline (PBS). MTT working solution in complete medium (final concentration 0.5 mg/mL) at 100 µL was added to each well and cells were incubated in an atmosphere humidified at 5%, with CO_2_ at 37 °C. After incubation for 4 h, the supernatant was removed, the dark blue of formazan crystal was dissolved in 100 µL dimethyl sulfoxide (DMSO; Sigma-Aldrich, St. Louis, MO, USA), and the plates were shaken for five minutes. The optical density of dissolved formazan crystal was read at 570 nm, with reference wavelength at 650 nm, using a SpectroMax M3 microplate reader (Molecular Devices, Sunnyvale, CA, USA).

### 4.5. RNA Extraction

The total RNA of each sample was extracted using RNeasy^®^ PlusMini kit (Qiagent, Hilden, Germany), and was qualified and quantified by Agilent 2100 Bioanalyzer (Agilent Technologies, Palo Alto, CA, USA), NanoDrop 2000 (Thermo Fisher Scientific Inc., Waltham, MA, USA) and 1% agarose gel. Total RNA at 1 µg with a RIN (RNA Integrity Number) value above seven was used for the following library preparation.

### 4.6. RNA Sequencing and Quality Control

Next generation sequencing library preparations were constructed according to the manufacturer’s protocol (NEBNext^®^Ultra^TM^ RNA Library Prep Kit for Illumina^®^). The poly(A) mRNA isolation was performed using NEBNext Poly(A) mRNA Magnetic Isolation Module (NEB)or Ribo-Zero rRNA removal Kit (Illumina, San Diego, CA, USA). The mRNA fragmentation and priming were then performed by using NEBNext First Strand Synthesis Reaction Buffer and NEBNext Random Primers. First strand cDNA was synthesized using ProtoScript II Reverse Transcriptase, and the second-strand cDNA were synthesized using Second Strand Synthesis Enzyme Mix. The purified double-stranded cDNA by AxyPrep Mag PCR Clean-up (Axygen Biosciences, Central Avenue, Union City, CA, USA) was then treated with End Prep Enzyme Mix to repair both ends and to add a dA-tailing in one reaction, followed by a T-A ligation to add adaptors to both ends. The size selection of Adaptor-ligated DNA was then operated by using the AxyPrep Mag PCR Clean-up (Axygen Biosciences, Central Avenue, Union City, CA, USA), and fragments of ~360 bp (with the approximate insert size of 300 bp) were recovered. Each sample was then amplified by PCR for 11 cycles, using P5 and P7 primers, with both primers carrying sequences which can anneal with flow cell to perform bridge PCR, and P7 primer, carrying a six-bases index allowing for multiplexing. The PCR products were cleaned up using AxyPrep Mag PCR Clean-up (Axygen Biosciences, Central Avenue, Union City, CA, USA), and validated using an Agilent 2100 Bioanalyzer (Agilent Technologies, Palo Alto, CA, USA), and quantified by Qubit 2.0 Fluorometer (Invitrogen, Carlsbad, CA, USA). Then, libraries with different indices were multiplexed and loaded on an Illumina HiSeq instrument according to manufacturer′s instruction (Illumina, San Diego, CA, USA). Sequencing was carried out using a 2 × 150 bp paired-end (PE) configuration; image analysis and base calling were conducted by the HiSeq Control Software (HCS) + OLB + GAPipeline-1.6 (Illumina, San Diego, CA, USA) on the HiSeq instrument. The sequences were processed and analyzed by Vishuo Biomedical, Singapore. In order to remove technical sequences, including adapters, polymerase chain reaction (PCR) primers, or fragments thereof, and quality of bases lower than 20, pass filter data of fasta format will be processed by Trimmomatic (v0.30) to obtain high quality clean data.

### 4.7. DEGs Identification from RNA-Seq

In the beginning, transcripts in fasta format were converted from known gff annotation files and indexed properly. Then, with the file as a reference gene file, HTSeq (v0.6.1) was estimated gene and isoform expression levels from the pair-end clean data. Differential expression analysis was conducted by using the DESeq Bioconductor package, a model based on the negative binomial distribution. After adjusted by Benjamini and Hochberg′s approach for controlling the false discovery rate, the *p*-value of genes was set at <0.05 to detect differential expressed genes [31]

### 4.8. Functional and Pathway Analysis

The DEGs from in silico and in vitro were combined and used for further integrated biological function analysis by using Venn diagram viewer. To investigate enriched functionally associated significant genes, Gene ontology (GO) analyses, including biological functions, molecular processes and cellular components, were performed using the Database for Annotation, Visualization and Integrated Discovery (DAVID; https://david.ncifcrf.gov/) version 6.8 [32]. The Kyoto Encyclopedia of Genes and Genomes (KEGG; http://www.genome.jp/kegg/) pathway analysis is an integrative data mining tool for the biological pathway interpretation of genome sequences and other high-throughput data. The pathway analysis was performed for enriched significant pathway [33]. A *p*-value <0.05 was set as the cut-off criterion.

### 4.9. Protein-Protein Interaction Network and Hub-Gene Identification

To explore proteins interaction networks of Differentially Expressed Genes (DEGs), a protein network was constructed by using NetworkAnalyst. NetworkAnalyst is an integrative online tool which is designed to support integrative gene expression analysis (https://www.networkanalyst.ca) [34]. The lists of DEGs were uploaded into the web-based server of NetworkAnalyst. Network construction was strictly set to contain only the original seed proteins. NetworkAnalyst provided betweenness centrality to measure the number through of node and node. The highest betweenness represents the critical point of the protein network.

### 4.10. TCGA Validation Using Combined DEGs

The mRNA expression of CCA (TCGA Provisional) was selected from biliary tract cancer (https://www.cbioportal.org/). Data were available for 51 samples. The total combined DEGs were applied for mRNA enrichment. The *p*-value cut-off was set at <0.05. The frequency dot plot was summarized by using cBioportal [35].

### 4.11. Immunohistochemistry Human Protein Atlas Validation Using Combined DEGs

The Human Protein Atlas contributes a large amount data on transcriptomics and proteomics in specific human tissue organs. The database is composed of a tissue atlas, a cell atlas and a pathology atlas (https://www.proteinatlas.org/) [36]. The combined DEGs and the differences in antibody-staining levels of cancer tissue samples and normal samples (cholangiocarcinoma VS. normal) were assessed, based on the available immunohistochemistry based staining levels in the HPA project.

### 4.12. Immunoblotting

At the end of chronic alcoholic treatment, the cells were lysed in lysis buffer containing 10 mM Tris (pH 7.4), 150 mM NaCl, 1% Triton X-100, 1 mM PMSF, 1 mM Na_3_VO_4_, 20 mM NaF and 1× protease inhibitor cocktail set I (Calbiochem, Germany). The cell lysates were then sonicated and incubated for 30 min at 4 °C before centrifuged 16,000× *g* for 15 min at 4 °C. The concentration of total protein was determined by using Bradford reagent (Bio-rad, Hercules, CA, USA). The 50 µg of protein was run onto a 7.5% SDS-polyacrylamide gel in a Mini-Protein II system (Bio-rad, Hercules, CA, USA). The separated protein was transferred onto nitrocellulose membrane using a Bio-rad mini Trans-Blot cell. The nitrocellulose membrane was incubated with blocking buffer containing 5% non-fat dry milk in a TBST buffer (10 mM Tris-HCl pH 8.0, 150 mM NaCl and 0.05% Tween-20) for 1 h at room temperature and followed by overnight incubation with primary antibodies at 4 °C. The antibodies against Cyclin D-1 (1:1000), β-actin (1:20,000) and MMP-2 (1:1500) were purchased from Cell Signaling Technology Inc. The membrane was washed three times with TBST for 10 min, and incubated with secondary antibodies conjugated with horseradish peroxidase (HRP) for 2 h. HRP conjugated secondary antibodies were as follow anti-rabbit IgG from Cell Signaling Technology Inc., anti-goat IgG from R&D system Inc, or anti-mouse IgG from Bio-Rad Laboratories, Inc. The protein bands were visualized using enhanced chemiluminescence (ECL) (GE Healthcare, UK). The intensity of bands was quantified by Image Quant TL software (GE Healthcare, UK).

### 4.13. Cell Doubling Time Determination

After chronic exposure to alcohol, the cells were seeded at 3 × 10^3^ cells on 6 wells plate culture dishes. After cell adherences, the cells were counted every 24 h for 6 days. The cells were trypsinized and stained with trypan blue. The viable cells were counted using Countess^®^ II FL Automated Cell counter (Thermo Fisher Scientific, UK). The doubling time was calculated from the cell group curve over 6 days using the following equation: Doubling time = (Final time − initial time) × (Log2/ log (final cell number) − log (initial cell number))

### 4.14. Wound Healing Assay

The motility of MMNK-1 after 60 days treatment with alcohol was assessed by using wound healing assay. The cells were seeded in 6 wells plate and grown to confluence. After 24 h, the complete medium was removed and replaced with serum-free medium to reduce the proliferative effect. Each well was scratched with a sterile pipette tip. After the incubation time at 0, 24, or 48 h, the serum-free medium was removed and washed suspended cells by PBS. The width of the denuded areas were measured. Wound closure was calculated as the percentage of the closed area of the initial width.

### 4.15. Statistical Analysis

Descriptive statistical analysis was generated for quantitative data and presented as mean ±SD. The mean ±SD. was calculated from 3 triplicates per sample. The difference between the 2 groups was determined by Student’s T test and *p*-value < 0.05, was considered significant.

## 5. Conclusions

This study is the first integrative transcriptomic analysis based on RNA-seq and computational bioinformatics, together with low dose of chronic alcohol exposure in human normal cholangiocyte in order to find an alcohol induced mechanism for CCA pathogenesis. The transcriptome data reveals that alcohol alters gene expression profiles in the human genome. A combination of in silico and in vitro experiments showed the biological process and molecular functions involved in the regulation of transcription from RNA polymerase II promoter; the cellular component is linked to nucleoplasm. KEGG pathway analysis also showed that chronic alcohol exposure is closely related to pathways in cancer. After prolonged alcohol exposure, oncogenic features, including cell doubling time determination and wound healing assay, were performed. Our results suggest that chronic alcohol exposure induced oncogenesis characteristics through CCND-1 and MMP-2 up-regulation, respectively. Finally, in clinical verification, *FOS* expression was significantly associated with alcohol-induced CCA pathogenesis and could be development as early biomarker. This study provides an understanding of a global transcriptional activity and confirms the possible oncogenic effects in human normal cholangiocytes with prolonged alcohol consumption behavior. However, further in vivo studies should be performed to validate the current in vitro and in silico results.

## Figures and Tables

**Figure 1 ijms-20-05987-f001:**
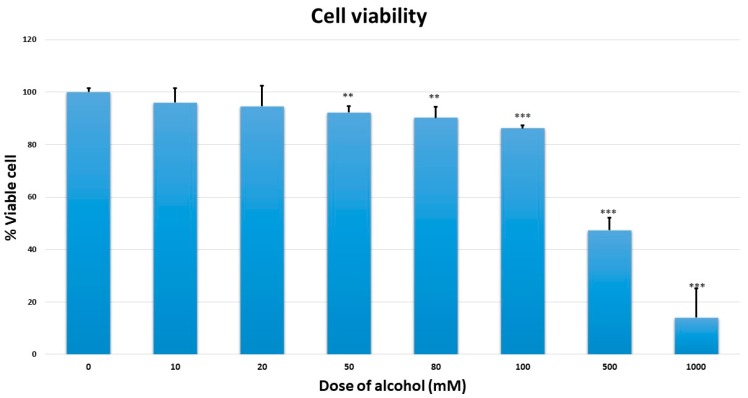
Cell viability assay of MMNK-1 cells exposed to a range of alcohol concentrations (0, 10, 20, 50, 80, 100, 500, and 1000 mM) for 7 days. “**” and “***” indicate significant differences (*p* < 0.05) and (*p* < 0.01), respectively.

**Figure 2 ijms-20-05987-f002:**
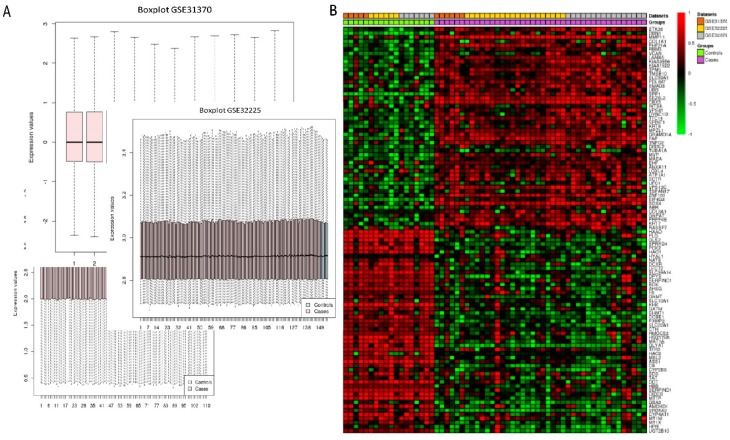
Box plot of data normalization and clustering heat map of 3 datasets, including GSE31370, GSE32225 and GSE32879. (**A**) Box plot of data normalization. The X-axis represents normal control and cholamgiocarcinoma samples and Y-axis represents gene expression value. (**B**) Hierarchical clustering heat map of DEGs from 3 datasets. Red indicates up-regulated genes and green indicates down-regulated genes.

**Figure 3 ijms-20-05987-f003:**
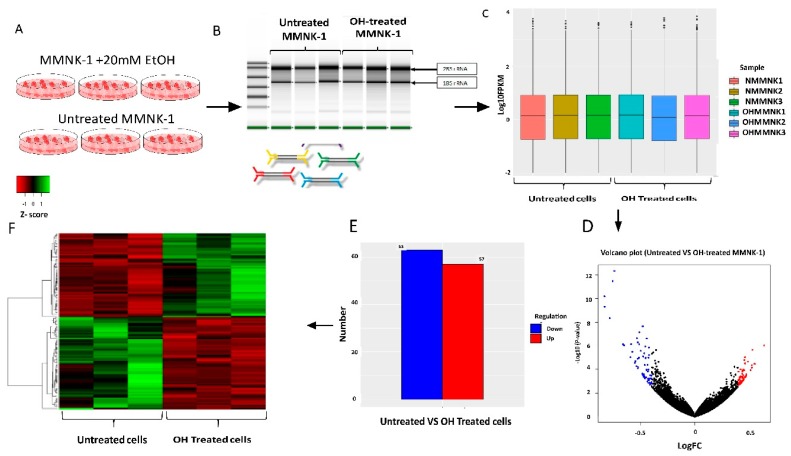
Overall experimental design of in vitro RNA-sequence. (**A**) MMNK-1 cells were maintained as a monolayer culture. Untreated and alcohol OH-treated MMNK-1 were exposed with 20 mM alcohol for 2 months and fresh media with and without alcohol were daily changed. (**B**) After chronic alcohol exposure, cells were collected and isolated total RNA for each sample. The cell processing was repeated 3 times. (**C**) Box plot of data normalization (**D**) The spots on volcano plot represent each gene with untreated and OH-treated MMNK-1. The red dot represents up-regulated genes; the blue dot represents down-regulated genes and the black dot represents genes which are not DEGs between groups. (**E**) The number of DEGs in transcriptomic comparison between 2 groups. (**F**) Hierarchical heat map of gene expression pattern. The red indicates up-regulated genes and the green indicates down-regulated genes.

**Figure 4 ijms-20-05987-f004:**
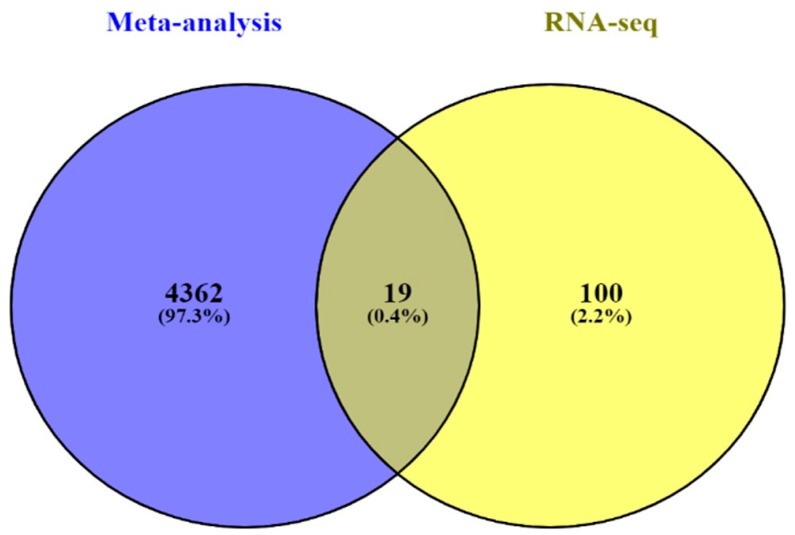
Venn diagram of combined DEGs between in silico meta-analysis of cholangiocarcinoma and in vitro RNA-Seq. The diagram reveals 19 overlapping genes.

**Figure 5 ijms-20-05987-f005:**
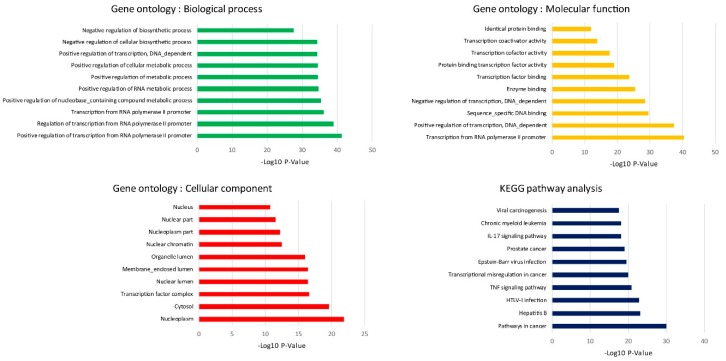
GO functional and KEGG pathway analysis of combined 19 DEGs. The Y-axis displays the categories of KEGG and GO including biological process, molecular function and cellular component.

**Figure 6 ijms-20-05987-f006:**
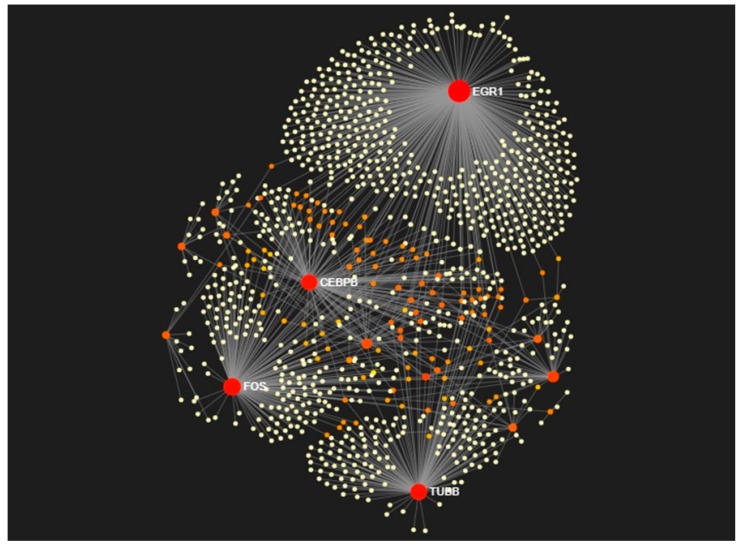
The PPI network of combined 19 DEGs. A total of 1058 nodes, 1226 edges and 17 seeds constitute the network. The hub-gene can be identified according to the degree of connectivity and is revealed by the red dot. The orange, yellow and white dots represent to the degree of connectivity that decreased accordingly.

**Figure 7 ijms-20-05987-f007:**
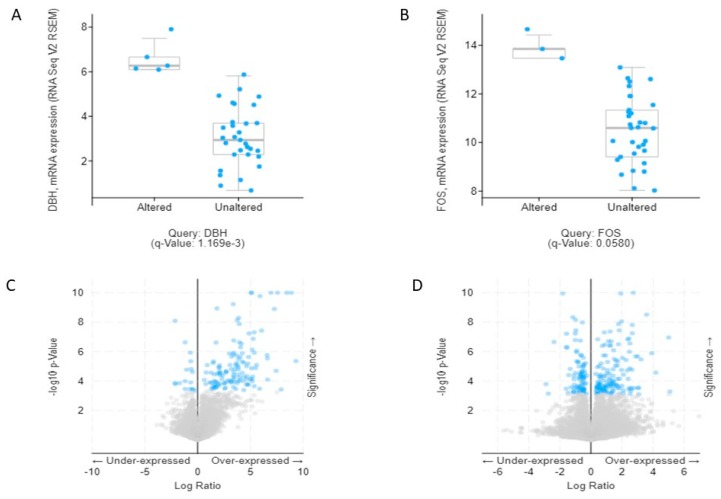
The mRNA expression analysis in cholangiocarcinoma (cBioportal). (**A**,**B**) The box plot comparing *DBH* and *FOS* gene expression in altered (left plot) and unaltered (right plot) groups were identified from cBiopotal. (**C**,**D**) Volcano plot of mRNA expression profile of *DBH* and *FOS*. The upper right dot represents up-regulation and the upper left dot represents down-regulation.

**Figure 8 ijms-20-05987-f008:**
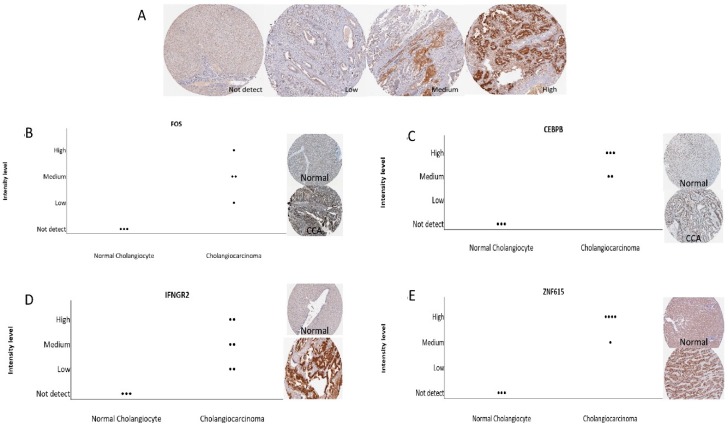
Validation of the combined 19 DEGs with immunohistochemistry from HPA database. (**A**) The differences of antibody-staining levels include not detected, low, medium and high. (**B**–**E**) CCA-specific genes including *FOS*, *CEBPB*, *INFGR2* and *ZNF615*. The frequency dot plots showed positive correlation with the expression of four genes with CCA tissue while none were observed in normal tissue.

**Figure 9 ijms-20-05987-f009:**
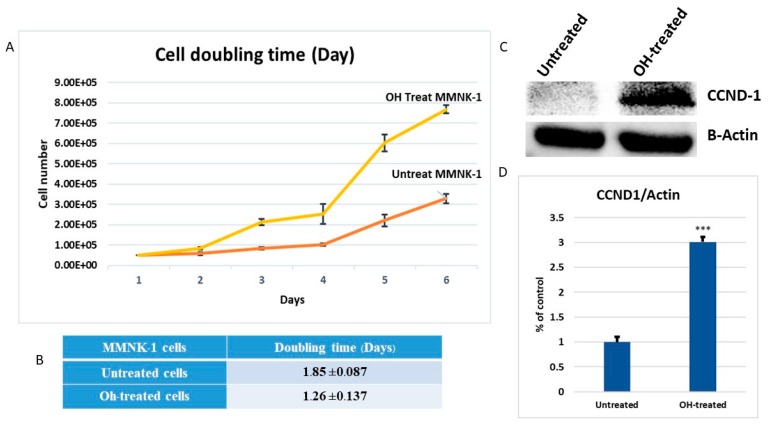
The doubling time of untreated and OH-treated MMNK-1 after chronic exposure and CCND-1 expression. (**A**,**B**) According to the proliferation curve, showed untreated MMNK-1’s higher rates of proliferation. (**C**) Western blot analysis for cell cycle and proliferation marker, CCND-1 was carried out with lysates from 3 replicated samples. (**D**) Quantification of the CCND-1 expression levels (intensity levels as a ratio of β-actin). ”***” indicates significant differences (*p* < 0.01).

**Figure 10 ijms-20-05987-f010:**
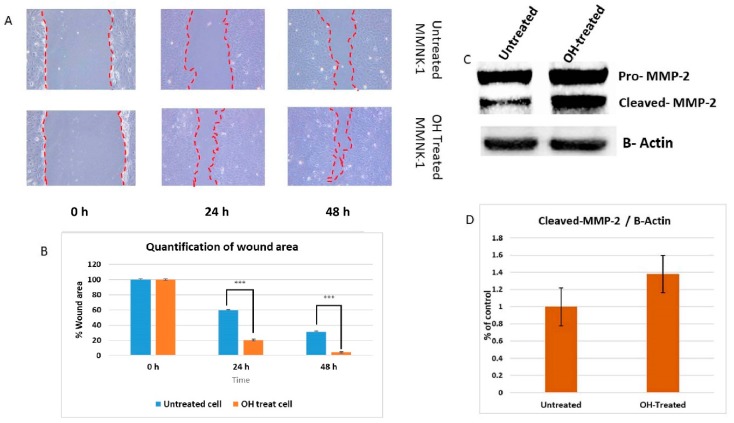
Wound healing assay and matrix metalloproteinase (MMP) 2 expression. (**A**,**B**) Wound healing assay using untreated and OH-treated MMNK-1 after chronic alcohol exposure and quantification of percentage wound area. (**C**) Relative MMP-2 expression in untreated and OH-treated MMNK-1 were measured by western blot analysis. (**D**) Quantification of the MMP-2 expression levels intensity using β-actin as a ratio. ”***” indicates significant differences (*p* < 0.01).

**Figure 11 ijms-20-05987-f011:**
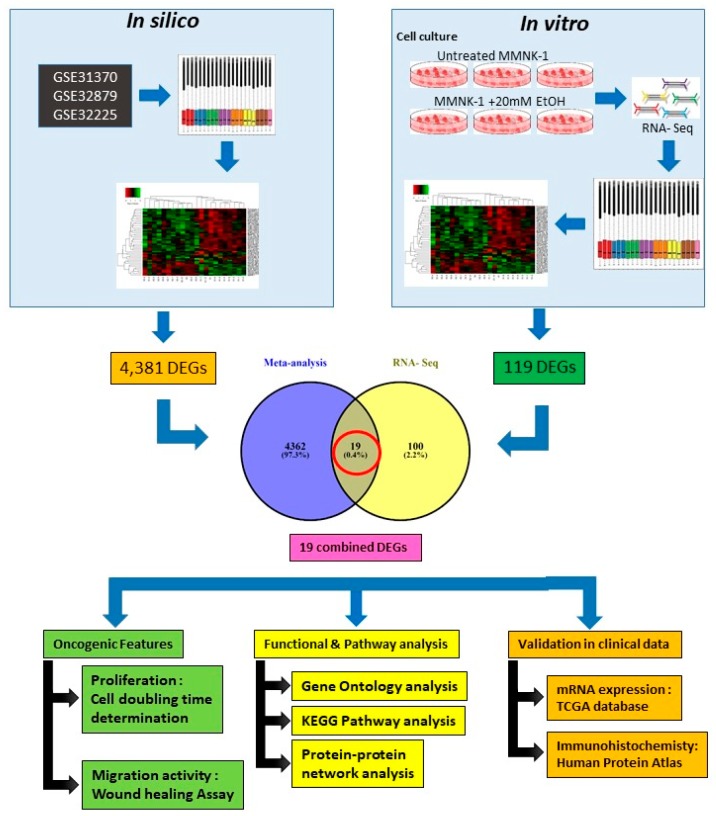
Overall experimental design in this study.

**Figure 12 ijms-20-05987-f012:**
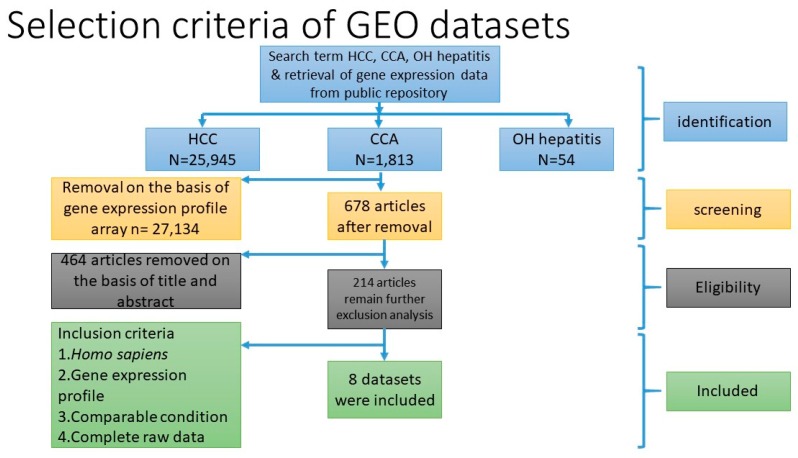
Flow diagram of the system review and meta-analysis results.

**Table 1 ijms-20-05987-t001:** The 19 genes list and description of combined 19 DEGs between in silico meta-analysis and in vitro RNA-Seq.

Gene Symbol	Description
*TRIM38*	Tripartite Motif Containing 38
*MC1R*	The melanocortin 1 receptor
*DBH*	Dopamine Beta-Hydroxylase
*TUBB*	Tubulin Beta Class I
*CD58*	CD58 Molecule
*FST*	Follistatin
*FOS*	Fos proto-oncogene
*IFNGR2*	interferon gamma receptor 2
*FHL1*	four and a half LIM domains 1
*EGR1*	early growth response 1
*DIO2*	iodothyronine deiodinase 2
*CEBPB*	CCAAT enhancer binding protein beta
*CTXN1*	Cortexin 1
*EDIL3*	EGF Like Repeats And Discoidin Domains 3
*CSF2*	Colony Stimulating Factor 2
*RGMB*	Repulsive Guidance Molecule BMP Co-Receptor B
*TRPC4*	Transient Receptor Potential Cation Channel Subfamily C Member 4
*ZNF615*	Zinc Finger Protein 615
*VGLL2*	Vestigial Like Family Member 2

**Table 2 ijms-20-05987-t002:** The most 12 hub-genes identified, based on PPI network analysis.

Hub-Gene	Description	Degree	Betweness
*EGR1*	Early Growth Response 1	518	389231.3
*FOS*	Fos Proto-Oncogene	214	161034.5
*TUBB*	Tubulin Beta Class I	173	141947.7
*CEBPB*	CCAAT Enhancer Binding Protein Beta	155	128867.6
*FHL1*	Four and a Half LIM Domains 1	45	33244.1
*CSF2*	Colony Stimulating Factor 2	31	17950.08
*TRIM38*	Tripartite Motif Containing 38	16	10085.46
*DIO2*	Iodothyronine Deiodinase 2	15	9810.154
*FST*	Follistatin	13	11562
*TRPC4*	Transient Receptor Potential Cation Channel Subfamily C Member 4	12	9559.997
*MC1R*	Melanocortin 1 receptor	12	11561
*UBC*	Ubiquitin C	11	33010.69

**Table 3 ijms-20-05987-t003:** The study information in in silico meta-analysis. (**A**) The basal information of 3 studies. (**B**) The significant genes in each study after gene expression analysis.

Study (A)	Platform	Samples	
		Normal	Disease
GSE31370	Illumina HumanHT-12 V4.0 expression beadchip	5	6
GSE32225	Illumina HumanRef-8 WG-DASL v3.0	6	149
GSE32879	Affymetrix Human Gene 1.0 ST Array	7	16
Total		18	171
**Study (B)**	**Significant Genes**		
GSE31370	116		
GSE32225	3169		
GSE32879	4997		
Meta-genes	4381		

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
