# Peer review of "Integrative In Silico and In Vitro Transcriptomics Analysis Revealed Gene Expression Changes and Oncogenic Features of Normal Cholangiocytes after Chronic Alcohol Exposure"

_ijms, 2019, doi:10.3390/ijms20235987_

Round 1

Reviewer 1 Report

The paper reports an interesting study about the transcriptomic analysis on the effects of alcohol exposure on normal cholangiocyte.

The authors used a combined in silico and in vitro approach and identified a panel of genes important in this correlation.

The results are interesting and the paper is acceptable for publication.

There is only a lacking point. The authors should perform a in vivo study to verify experimentally our in vitro amd in silico results. The authors should add a comment about this point in the conclusion section.

Author Response

Response to review comments

Ms. Ref. No.:        ijms-635993

Ms. Title:              Integrative in silico and in vitro transcriptomics analysis

                              revealedcgene expression changes and oncogenic features of

                              chronic alcohol exposure to normal cholangiocyte

Corresponding Author:  Jutamaad Satayavivad, Ph.D.

Journal:                International Journal of Molecular Sciences

The authors would like to thank the editor and reviewers very much for their valuable time for reviewing this manuscript.  We are also grateful for their constructive and important comments/suggestions to improve the quality of this manuscript. We hope that our revised manuscript has been improved to a level of reviewers’ satisfaction. 

Overall, the authors attempted to revise the manuscript according to the points that suggested by the reviewers. 

***************************************************************************

Reviewer 1
comments and Suggestions for Authors

The paper reports an interesting study about the transcriptomic analysis on the effects of alcohol exposure on normal cholangiocyte.

The authors used a combined in silico and in vitro approach and identified a panel of genes important in this correlation.

The results are interesting and the paper is acceptable for publication.

There is only a lacking point. The authors should perform a in vivo study to verify experimentally our in vitro and in silico results. The authors should add a comment about this point in the conclusion section.

Response 1;

          We are thankful for your valuable comment. We agree with the reviewer that an in vivo study should be further performed to validate the in vitro and in silico results.  Based on your comments, the sentence “However, further in vivo studies should be performed to validate the current in vitro and in silico results” has been added at the end of discussion and conclusion sections. (page 11, line 411-412)

Reviewer 2 Report

In the manuscript titled, "Integrative in silico and in vitro transcriptomics analysis revealed gene expression changes and oncogenic features of chronic alcohol exposure to normal cholangiocyte" by Chujan et al., the authors present correlative gene expression changes in cholangiocytes and CCA and associate 19 gene changes to chronic exposure to alcohol. 

It is very obvious that the authors have a strong background in working with big data sets and used a myriad of bioinformatic analyses to test their hypothesis. The manuscript is well written and the data is very supportive of their conclusions for the most part. There are two major flaws of the paper they need to be corrected before making it publicly available. 

Major flaw #1

In section 3.9 of the results, the authors state they are testing the invasive nature of MMNK-1 cells with and without chronic exposure to alcohol. While they immunoblotted for a classical invasion-associated protein (MMP-2), the scratch assay is not a test of invasiveness. The scratch assay is controversial in whether it is testing migration or proliferation to close the scratched surface area, but it is definitely not an invasion assay. If the authors would like to explore the invasion assay, they should consider doing a matrigel or chamber invasion assay. Alternatively, the authors could rewrite the section and remove any references of invasion throughout the paper. 

Major flaw #2

Risk factors are not causations for CCA. This came out in the Discussion section and few other places. The environmental factors that associate with an increased risk are just that, associations and correlations. The authors need to correct this overstatement throughout the manuscript. 

Minor things to address: 

The paper would have more substance if the authors took one of the genes of interest and interrogated its interactions further by genetic or pharmacological manipulation. For example the CCND1, MMP2, or FOS would have been great candidates to further explore and confirm their contributions to alcohol induced/enhanced transformation. It felt like a lot of shallow off shoots. 

The sentence on line 37 in the introduction needs to be reworded. It is unclear how the different things are tied DNA damage and alcohol consumption. 

The methods section discussing the cell viability experiments needs further clarification about the volumes or final concentration of MTT and DMSO used. 

Also in the methods section under immunoblotting it is not expressly mentioned which secondary antibody was used. 

Lastly, there are no error bars on the line graphs for the proliferation assay. 

Reviewer 3 Report

The manuscript presented by Dr Suthipong Chujan and colleagues is focused on the analysis of the link between chronic alcohol exposure and cholangiocarcinogenesis. The authors use for their goal the immortalized human cholangiocyte cells MMNK-1.

All the bioinformatics analysis are well-performed, also considering the graphical art that is very good.  The studies of the comparison between the differentially expressed genes are good, but the biological deepening is too much preliminary. 

The pictures are not located in the text as usually happens in IJMS.

My Questions are:

-In lines 89-90 the authors write "The media with ethanol were replaced every 24 h with fresh media containing the 20mM ethanol concentration. The cells were treated for 60 days (2 months) and trypsinized 80% twice weekly when archived. After 60 days, the cells were used for further experiments". In this section is not well explained why the authors make this long treatment. Probably they should divide this point, cause it is a central point for understanding the results section. 

-In wound healing assay the authors should add an inhibitor of proliferation. 

-Is Paragraph 3.1  supported by a figure? In the text is not written. 

-Bioinformatics studies are not completely new data. They are already present in databases and the authors only collected or interpreted or re-used the data. At this purpose, also if they are well performed the manuscript still lacks of an explanation of the biological effects.

-In paragraph 3.8 the authors should use also another method for demonstrating their result.

-In 3.9 paragraph with wound healing the authors are not observing invasion of the cells, but migration. For demonstrating their goal and for showing cell migration or invasiveness the authors should use Boyden's chambers coated with collagen or matrigel. For a hint, they should follow materials and methods in  https://doi.org/10.3390/cancers11060784. 

-Furthermore in the Results section is not very clear how they reveal MMP.

Metalloproteases are secreted proteins. Authors should use conditioned medium and perform zymography or western blot after the concentration of the medium. Another way for observing if their cells have the capability of disrupting the extracellular matrix is the use of a commercially available fluorescent matrix.

Author Response

Response to review comments

Ms. Ref. No.:        ijms-635993

Ms. Title:              Integrative in silico and in vitro transcriptomics analysis

                              revealedcgene expression changes and oncogenic features of

                              chronic alcohol exposure to normal cholangiocyte

Corresponding Author:  Jutamaad Satayavivad, Ph.D.

Journal:                International Journal of Molecular Sciences

The authors would like to thank the editor and reviewers very much for their valuable time for reviewing this manuscript.  We are also grateful for their constructive and important comments/suggestions to improve the quality of this manuscript. We hope that our revised manuscript has been improved to a level of reviewers’ satisfaction. 

Overall, the authors attempted to revise the manuscript according to the points that suggested by the reviewers. 

***************************************************************************

Reviewer 3

Comments and Suggestions for Authors

The manuscript presented by Dr Suthipong Chujan and colleagues is focused on the analysis of the link between chronic alcohol exposure and cholangiocarcinogenesis. The authors use for their goal the immortalized human cholangiocyte cells MMNK-1.

All the bioinformatics analysis are well-performed, also considering the graphical art that is very good. The studies of the comparison between the differentially expressed genes are good, but the biological deepening is too much preliminary.

The pictures are not located in the text as usually happens in IJMS.

Response 1;

We thank the reviewer for the careful and critical reading of our manuscript and gave a number of very useful comments and suggestions. Our detailed answers are given below.

We uploaded the attachment of the figures separately due to there are a lot of figures in our manuscript. We are very sorry for that.

My Questions are:

-In lines 89-90 the authors write "The media with ethanol were replaced every 24 h with fresh media containing the 20mM ethanol concentration. The cells were treated for 60 days (2 months) and trypsinized 80% twice weekly when archived. After 60 days, the cells were used for further experiments". In this section is not well explained why the authors make this long treatment. Probably they should divide this point, cause it is a central point for understanding the results section.

Response 2;

We chose alcohol treatment concentration at the 20mM concentration to represent chronic alcohol exposure as a previous study described where the chronic alcohol treatment at 20 mM for 9 days could alter cellular functionality of rat astrocyte cell culture [12]. For information, the 20 mM in vitro alcohol concentration approximates a 100 mg/dL blood alcohol level, which is achieved in vivo after a dose of moderate drink and is the upper range of blood alcohol concentration in most United State jurisdiction [13]. The duration of long-term alcohol exposure for 60 days was referred from previous established long-term alcohol exposure model [14].  We really thankful for the recommendation and the clear context has now been added on the line 87- 92.

-In wound healing assay the authors should add an inhibitor of proliferation.

Response 3;

Thank you for the comment. In the section of wound healing assay, we performed the cell culture without fetal bovine serum or serum free medium for reducing the proliferative effect. To be clear context, the sentence “. After 24 h, the complete medium was removed and replaced with serum-free medium to reduce the proliferative effect” has been added in this section on page 6, line 203. However, your comment is very useful for our further works, and is important point to study the downstream pathway cellular signaling of alcohol.

-Is Paragraph 3.1 supported by a figure? In the text is not written.

Response 4;

The word “Figure 4” has already appeared on the line 206.

-Bioinformatics studies are not completely new data. They are already present in databases and the authors only collected or interpreted or re-used the data. At this purpose, also if they are well performed the manuscript still lacks of an explanation of the biological effects.

Response 5;

Thank you for the suggestion, in this study, we do not only perform just the in silico bioinformatics from publicly repository data, and we also combined the transcriptomics data from RNA- seq with the in silico. In this purpose, we would like to see the combination of differentially expressed genes (DEGs) and observe the effects of prolong alcohol exposure toward cholangiocarcinoma development. In this step, we received the genes from both two sources, in silico and in vitro for biological function analysis.

For biological effects determination, we performed the analyses by using gene ontology and KEGG pathway analysis. The results show various biological alterations and it has already explained in the section of discussion. However, the –omics analysis is generally screening a global alteration. Any points of –omics analysis may not be all explained. We just focus on the interesting points or previous evidences that correlate and associate to our purposes and mention it for an introduction of the further works.

-In paragraph 3.8 the authors should use also another method for demonstrating their result.

Response 6;

Thank you very much for this comment, with respect to reviewer, measuring cell proliferation can be performed by various different methods, each with varying levels of sensitivity, reproducibility and compatibility with high-throughput formatting.

          According to our study, the aim of the present study is to determine a cell scale observable, namely the doubling time determination, after prolong treatment. In addition, we also use another method to verify our results using western blotting and the result showed the positive correlation with the CCND-1 expression.

          Recently, cell population doubling time have been used in the study for determining of cell proliferation, so we thought that it should be appropriated to use this method which can suggest the abnormality in cell growth.

reference

Zhan, X.-S.; El-Ashram, S.; Luo, D.-Z.; Luo, H.-N.; Wang, B.-Y.; Chen, S.-F.; Bai, Y.-S.; Chen, Z.-S.; Liu, C.-Y.; Ji, H.-Q. A Comparative Study of Biological Characteristics and Transcriptome Profiles of Mesenchymal Stem Cells from Different Canine Tissues. Int. J. Mol. Sci. 2019, 20, 1485.

-In 3.9 paragraph with wound healing the authors are not observing invasion of the cells, but migration. For demonstrating their goal and for showing cell migration or invasiveness the authors should use Boyden's chambers coated with collagen or matrigel. For a hint, they should follow materials and methods in https://doi.org/10.3390/cancers11060784.

Response 7;

We also agree with this comment. The contexts about invasion have been reworded all the revised manuscript for cell motility and migration.

-Furthermore in the Results section is not very clear how they reveal MMP.

Metalloproteases are secreted proteins. Authors should use conditioned medium and perform zymography or western blot after the concentration of the medium. Another way for observing if their cells have the capability of disrupting the extracellular matrix is the use of a commercially available fluorescent matrix.

Response 8;

MMP-2 has been reported to play a critical role in cancer cell migration and invasion by contributing to the degradation of the ECM and cancer progression [1]. The previous study showed that expression of MMP-2 has an important role in tumorigenesis of biliary tract cancer [2,3].

According to your comment, we agree that method would be appropriated to study the enzymatic activity of MMP-2. As below uncropped figure (please see the attached file), there are two bands of MMP-2 expression including pro MMP-2 (72 KDa) and cleaved MMP-2 (64 KDa) that represents to the activated form of MMP-2. The cleaved MMP-2 is increased according to the alcohol- treated cells and the result from migration assay. Based on our facility and previous study, we think that although the western blotting may not be the best mothed for MMP-2 detection but it could be used for this study [4]. Thank you for a very good suggestion.

References

[1] verall, C.M.; Dean, R.A. Degradomics: Systems biology of the protease web. Pleiotropic roles of MMPs in cancer. Cancer Metastasis Rev. 2006, 25, 69–75.

[2] Braicu, C. Molecular Markers in the Pathogenesis of Cholangiocarcinoma: Potential for Early Detection and Selection of Appropriate Treatment. Gastroenterology Research 2009.

[3] Karadag, N.; Kirimlioglu, H.; Isik, B.; Yilmaz, S.; Kirimlioglu, V. Expression of Matrix Metalloproteinases in Gallbladder Carcinoma and Their Significance in Carcinogenesis. Applied Immunohistochemistry & Molecular Morphology 2008, 16, 148–152.

[4] Ricci, C.; Mota, C.; Moscato, S.; D’Alessandro, D.; Ugel, S.; Sartoris, S.; Bronte, V.; Boggi, U.; Campani, D.; Funel, N.; Moroni, L.; Danti, S. Interfacing polymeric scaffolds with primary pancreatic ductal adenocarcinoma cells to develop 3D cancer models. Biomatter 2014, 4.

Round 2

Reviewer 3 Report

The manuscript is improved in all the sections in this form.

A lot of details have been added by the authors. 

In the figures showed, there is only a graphical art discrepancy. For example, the pictures about bioniformatic data are very beautiful, graphs about proliferation or pictures in which are western blots are not at the same level of graphical art.

The complete western blot of MMP that are now added in the present form of the manuscript clarify better their goal. 

Author Response

Response to review comments (Round 2)

Ms. Ref. No.:        ijms-635993

Ms. Title:              Integrative in silico and in vitro transcriptomics analysis

                              revealedcgene expression changes and oncogenic features of

                              chronic alcohol exposure to normal cholangiocyte

Corresponding Author:  Jutamaad Satayavivad, Ph.D.

Journal:                International Journal of Molecular Sciences

Reviewer 3 (Round 2)

Comments and Suggestions for Authors

The manuscript is improved in all the sections in this form.

A lot of details have been added by the authors.

In the figures showed, there is only a graphical art discrepancy. For example, the pictures about bioniformatic data are very beautiful, graphs about proliferation or pictures in which are western blots are not at the same level of graphical art.

The complete western blot of MMP that are now added in the present form of the manuscript clarify better their goal.

Response 1;

          We are thankful for your recommendation. The figure 4, 11, and 12 which including graphs and western blots have been edited to provide a better quality.